# Synergistic catalysis for cascade allylation and 2-aza-cope rearrangement of azomethine ylides

Liang Wei[1], Qiao Zhu[1], Lu Xiao[1], Hai-Yan Tao[1] & Chun-Jiang Wang[1,2]

The efficient construction of enantiomerically enriched molecules from simple starting materials via catalytic asymmetric synthesis strategies is a key challenge in synthetic chemistry. Metallated azomethine ylides are commonly-used synthons for the preparation of N-heterocycles and α-amino acids. Remarkably, to date, the utilization of azomethine ylides for the facile access to chiral amines has proven elusive. Here, we report that a synergistic Cu/Ir-catalytic system combined with careful tuning of the steric congestion can be used to convert aldimine esters to a variety of chiral homoallylic amines via a cascade allylation/2-aza-Cope rearrangement. The elucidation of the distinct effects of each stereogenic center of the allylation intermediates on the stereochemical outcome and chirality transfer in the rearrangement further guided the selection of catalysts combination.

---

[1] College of Chemistry and Molecular Sciences, Wuhan University, Wuhan 430072, China. [2] State Key Laboratory of Organometallic Chemistry, Shanghai Institute of Organic Chemistry, Shanghai 230021, China. Correspondence and requests for materials should be addressed to C.-J.W. (email: cjwang@whu.edu.cn)

The catalytic asymmetric construction of biologically important enantioenriched molecules from easily accessible and inexpensive starting materials is one of the long-term and formidable tasks in organic synthesis and medicinal chemistry[1,2]. Metal-stabilized azomethine ylides that are in situ-generated from aldimine esters, which could be formed via the simple condensation between aldehydes and α-amino acid esters, are particularly useful building blocks for the synthesis of therapeutic agents, clinically useful natural products, and precursors of other valuable organic compounds[3]. In general, structurally rigid metallated azomethine ylides are commonly-used 1,3-dipoles in cycloaddition reactions, and substantial effort has been directed toward the preparation of enantioenriched N-containing heterocycles (Fig. 1a)[4–7]. Alternatively, various nonproteinogenic chiral α-amino acids can be obtained via catalytic asymmetric Michael additions and allylation reactions, in which metallated azomethine ylides behave as viable nucleophiles (Fig. 1a)[8–15]. It is well known that the asymmetric attack of carbon-based nucleophiles to various imines (C=N) constitutes one of the most powerful methods for the construction of biologically important chiral amine derivatives[6]. For example, chiral homoallylic amines are prominently featured in numerous pharmaceuticals, agrochemicals, and building blocks in organic synthesis[16–20]. As a result, highly efficient approaches for the preparation of enantioenriched homoallylic amines have been studied extensively. The asymmetric nucleophilic addition of allylmetal reagents to imines represents the most well-established strategy to synthesize chiral homoallylic amines[18–20]. However, the use of moisture-sensitive allylmetal reagents, the required removal of protecting/activating group to give arguably more synthetically useful primary amine sometimes restrict its further application. On the other hand, there are no reports on the asymmetric

transformations of metallated azomethine ylides or their precursor aldimine esters that allow the preparation of chiral amines in a catalytic manner, although one example of highly diastereoselective addition of allyl organometal reagents to aldimine esters derived from (S)-valine was developed for the preparation of optically active homoallylic amine derivatives along with the aforementioned drawbacks[21].

Previously, we[11] and others[15] have reported Cu/Ir dual catalysis system for the stereodivergent construction[22,23] of α,α-disubstituted α-amino acids by the asymmetric allylation of aldimine esters with full control of the diastereoselectivity and enantioselectivity. The idea for the current approach to the synthesis of chiral homoallylic amines stemmed from the observation that an allyl-azaallyl core structure was imbedded in the branched allylation products. This led us to conjecture that such cross adducts obtained in our previous work[11] might further undergo a 2-aza-Cope rearrangement[24–38], leading to optically active homoallylic amines (Fig. 1b). Most recently, Niu and co-workers reported an elegant Ir-catalyzed asymmetric allylation of sterically bulky N-fluorenyl imines followed by 2-aza-Cope rearrangement to prepare enantioenriched 1,4-disubstituted homoallylic amines[39]. Therefore, we reasoned that enhancing the steric congestion of the adjacent stereogenic centers could break the stability of the initially formed allylation intermediates, which would provide a sufficient driving force for the subsequent 2-aza-Cope rearrangement via the release of the steric hindrance, pushing the equilibrium of the reaction toward the thermodynamically more stable product.

However, two problems had to be considered in the design of this transformation: (1) an appropriate substituent, which can enhance the steric congestion to promote the 2-aza-Cope rearrangement but would not suppress or deteriorate the stereo-

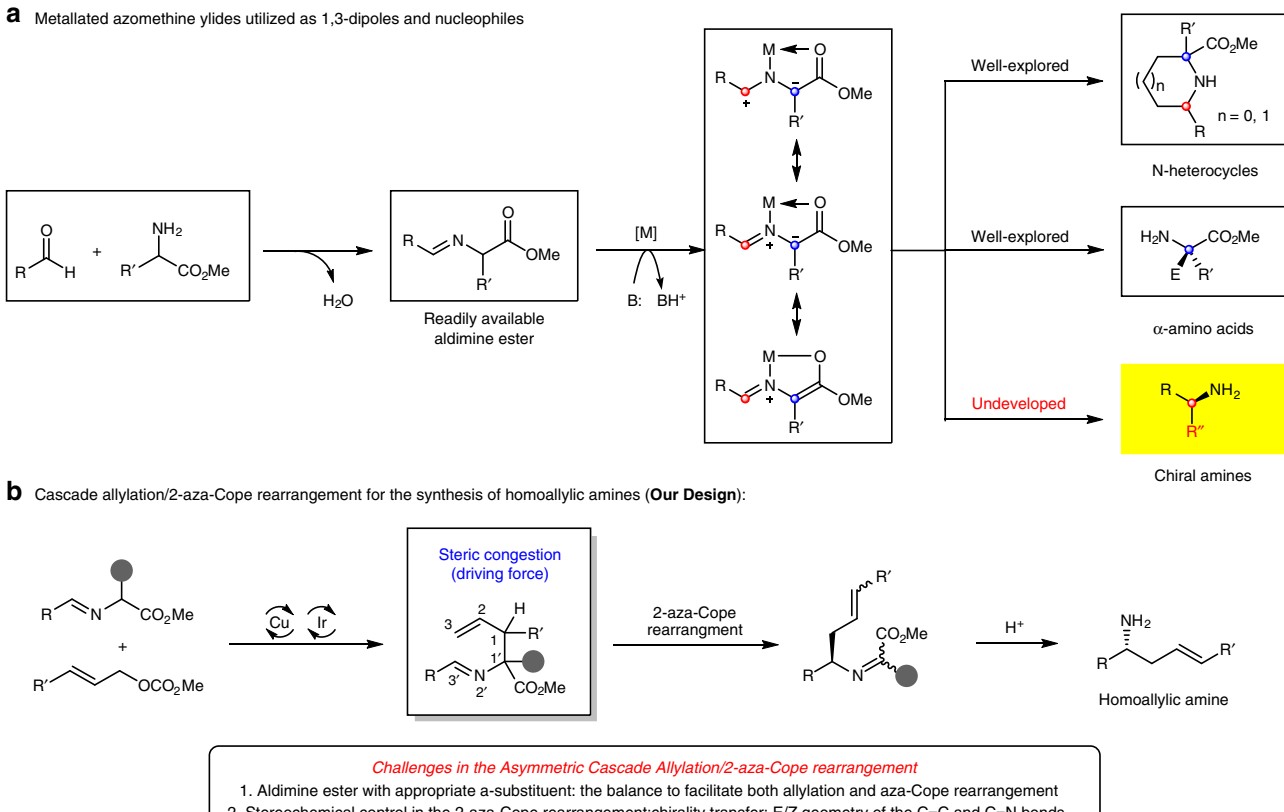

**a** Metallated azomethine ylides utilized as 1,3-dipoles and nucleophiles

**b** Cascade allylation/2-aza-Cope rearrangement for the synthesis of homoallylic amines (**Our Design**):

*Challenges in the Asymmetric Cascade Allylation/2-aza-Cope rearrangement*
1. Aldimine ester with appropriate a-substituent: the balance to facilitate both allylation and aza-Cope rearrangement
2. Stereochemical control in the 2-aza-Cope rearrangement:chirality transfer; E/Z geometry of the C=C and C=N bonds

**Fig. 1** Metallated azomethine ylide and its synthetic potential in organic synthesis. **a** Well-established and undeveloped synthetic applications of metallated azomethine ylides. **b** The approach developed in this study

| Entry[a] | R | Allylation | 2-aza-Cope | Yield (%)[b] | ee (%)[c] |
|---|---|---|---|---|---|
| 1 | *tert*-Butyl (**2a**) | ✕ | - | - | - |
| 2 | *iso*-Propyl (**2b**) | ✕ | - | - | - |
| 3 | *iso*-Butyl (**2c**) | ✓ | ✓ | 86 | 96 |
| 4 | Phenyl (**2d**) | ✓ | ✓ | 83 | 95 |

[a]All reactions were carried out with 0.20 mmol of **1a** and 0.30 mmol of **2** in 1 mL of CH$_2$Cl$_2$.
[b]Isolated yield. [c]Ee was determined by HPLC analysis.

**Fig. 2** Steric effect in this reaction. Initial test to find an aldimine ester bearing an appropriate α-substituent to facilitate both allylation and 2-aza-Cope rearrangement

integrity of the initial allylation, should be installed at the α-position of the aldimine ester; and (2) the efficacy of the chirality transfer and the E/Z-geometry control, which is rendered by the complicated stereochemistry of the 2-aza-Cope rearrangement, should be clarified.

Here, we show the development, the stereochemical control, the substrate scope generality and synthetic applications of the synergistic Cu/Ir-catalyzed asymmetric cascade allylation/aza-Cope rearrangement of azomethine ylides with allylic carbonates[40–44]. The current protocol was found to be a flexible and general process for the efficient preparation of a variety of 1,4-disubstituted homoallylic primary amines. Remarkably, the high efficiency of the synergistic Cu/Ir catalytic system was further validated by the expedient access to 1,3-disubstituted homoallylic amines which has seldom been obtained through catalytic asymmetric synthesis except sparse reports with the use of ally-metal reagents[45–48].

## Results

**Reaction development and optimization.** Initially, with cinnamyl methyl carbonate **1a** as the π-allyl precursors, the reactivity of several bulky α-substituted aldimine esters **2a**–**2d** were investigated under our previously reported dual Cu/Ir catalyzed allylation reaction conditions[11] in order to find the balance between the allylation and the aza-Cope rearrangement. Both tert-butyl and iso-propyl groups were determined to be too bulky, and corresponding aldimine esters **2a** and **2b** from (±)-*tert*-leucine and (±)-valine were nonreactive in the allylation catalyzed by the set of catalyst combinations [Cu(I)/(S,S$_p$)-**L1** + Ir(I)/(S,S,S)-**L2**], which we attribute to the reduced nucleophilicity of the Cu-coordinated azomethine ylide towards the in situ-formed electrophilic Ir-π-allyl intermediate (Fig. 2, entries 1 and 2). Considering the overcorrection of the steric hindrance, we inferred that replacing the more sterically hindered tert-butyl or iso-propyl group in the aldimine esters with a less sterically demanded group might allow a proper balance by promoting the 2-aza-Cope rearrangement through the required steric congestion while still maintaining sufficient nucleophilicity to facilitate the allylation process. When (±)-leucine derived imine ester **2c** was subjected to the dual catalytic system, the cascade allylation/2-aza-Cope rearrangement process occurred smoothly at room temperature, and the desired homoallylic amine **3a**, with exclusively the E-geometrical olefin, was isolated in 86% yield and 96% ee via further acidic hydrolysis followed by protection with Boc$_2$O (Fig. 2, entry 3). (±)-2-Phenylglycine derived imine ester **2d** was also proved to be a viable nucleophilic precursor in this cascade reaction (Fig. 2, entry 4).

As the proper α-substituted aldimine esters were found to facilitate the cascade reaction, we further investigate the issue of the stereochemical control and chirality transfer in the 2-aza-Cope rearrangement. In our previous work, all four stereoisomers of the allylation products, which bear two adjacent stereogenic centers, could be predictably accessed by simply switching the dual Cu/Ir catalyst permutations because the two distinct metal catalysts exert almost absolute control over the corresponding stereogenic centers[11]. This stereodivergent catalytic system offers us an opportunity to study the roles of the two adjacent stereogenic centers generated in the allylation intermediates in the subsequent 2-aza-Cope rearrangement process in great detail. Figure 3a–d illustrates the defining stereochemical feature of the cascade allylation/2-aza-Cope rearrangement with cinnamyl methyl carbonate **1a** and aldimine ester **2c** as the substrates. (R,E)-**3a** was separated in good yield with excellent enantioselectivity with catalyst combinations [Cu(I)/(S,S$_p$)-**L1** + Ir(I)/(R,R,R)-**L2**] or [Cu(I)/(R,R$_p$)-**L1** + Ir(I)/(R,R,R)-**L2**], and the corresponding allylation intermediates corresponded to (2S,3R)-**Int-3** or (2R,3R)-**Int-3**, respectively (Fig. 3a, b). (S,E)-**3a** was obtained with the other set of catalyst combinations [Cu(I)/(S,S$_p$)-**L1** + Ir(I)/(S,S,S)-**L2**] or [Cu(I)/(R,R$_p$)-**L1** + Ir(I)/(S,S,S)-**L2**], and the corresponding allylation intermediates corresponded to (2S,3S)-**Int-3** or (2R,3S)-**Int-3**, respectively (Fig. 3c, d). According to the results of these parallel experiments and our previous work, it is concluded that the absolute configuration of the stereogenic center in product **3a** is stereospecifically controlled by the tertiary allylic stereogenic center in the branched allylation intermediate **Int-3**, the late of which is exclusively controlled by the absolute configuration of chiral ligand **L2** in the corresponding set of Cu/Ir catalyst combination.

To gain more stereochemical insights, we attempted to isolate the rearranged imine product without additional work up. After several unsuccessful attempts with cinnamyl methyl carbonate **1a** as the π-allyl precursor, we finally realized the clean isolation of the rearranged imine product with 3-(6-methoxypyridin-3-yl) allyl methyl carbonate **1q** as π-allyl precursor (the corresponding imine products with more polar pyridine moiety are easier to be purified by silica gel chromatography). It was found that (S,E,E)-**4q**, containing an E-geometry of imine moiety, which was confirmed by NOESY experiments (see Supplementary Methods, Section Mechanism Study for details), was separated as the major isomer from the reaction with the set of catalyst combination [Cu(I)/(R,R$_p$)-**L1** + Ir(I)/(S,S,S)-**L2**] (Fig. 3e). In contrast, (S,E,Z)-**4q**, containing Z-geometry of imine moiety, was produced as the major isomer with the set of catalyst combination [Cu(I)/(S,S$_p$)-**L1** + Ir(I)/(S,S,S)-**L2**] (Fig. 3f). It is worth noting that according to the NMR experiments, the high ratio of the E/Z isomers

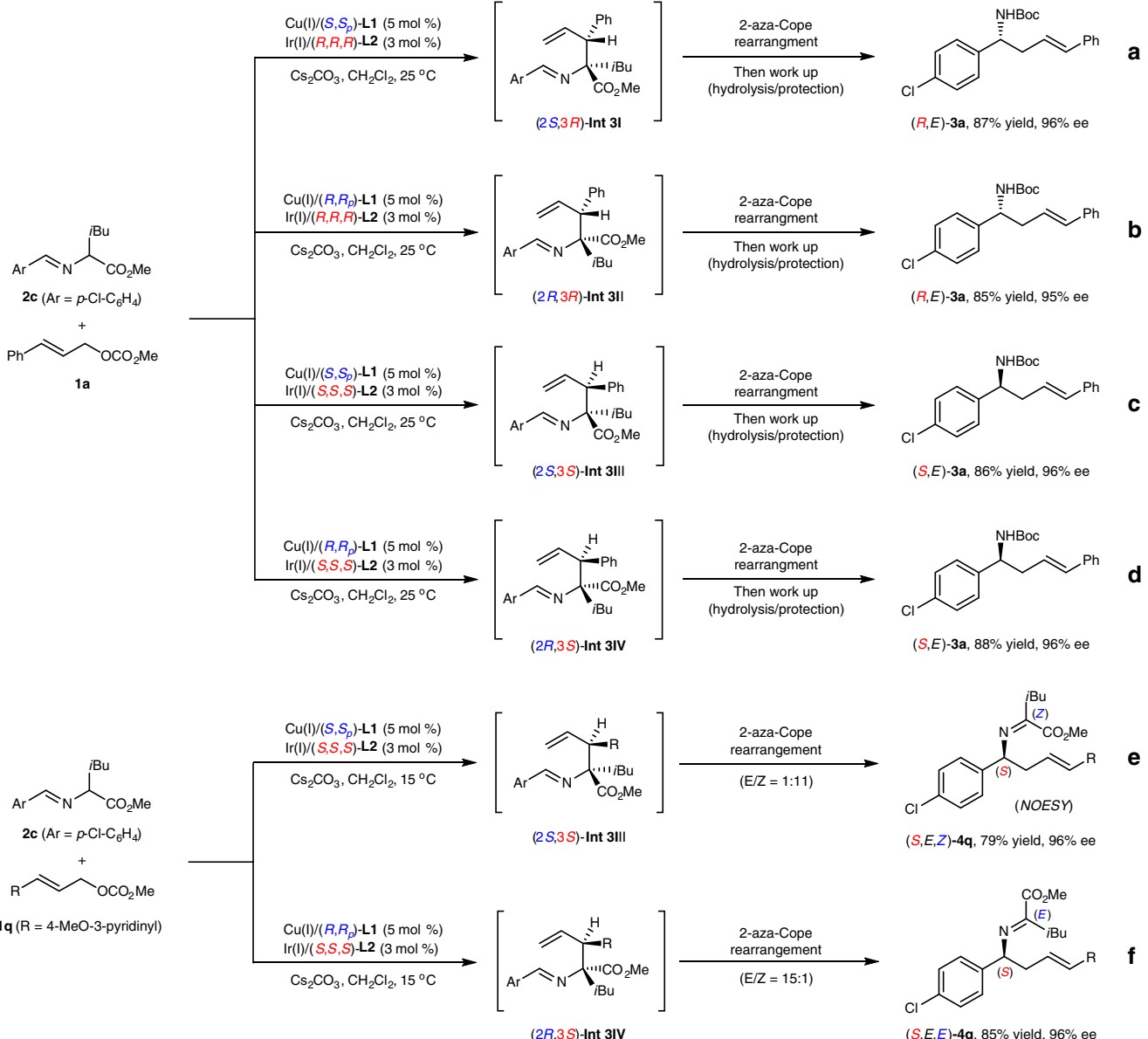

**Fig. 3** Parallel experiments with different set of catalyst combinations. **a–d** Study on the chirality transfer. **e–f** Study on the geometry of the generated imine moiety

gradually deteriorated into an equilibration with an almost 1:1 ratio via isomerization after standing for 3 days. Both isomers were successfully converted to (*S,E*)-**3q** with high enantioselectivity following the above-mentioned work up, which further confirmed that the two isolated imines **4q** were *E* and *Z* geometrical isomers of the C=N bond.

Based on the DFT calculations reported for the coordination of the azomethine ylide with Cu/(*S,Sp*)-**L1**[49] and the X-ray structure of intermediate π-ally-Ir/(*R,R,R*)-**L2**[50,51], the stereoselective allylation is controlled by the rigid geometry and facial selectivity of the formed Cu(I)-ylide as well as the approach of the formed Ir-π-allyl intermediate, and the stereodivergent protocol is attributed to the two distinct metal complex independently exerting almost absolute control over the corresponding stereogenic centers (Fig. 4). Therefore, the addition of the *Si*-face of the ylide species to the *Si*-face of the Ir-π-allyl species (*Si–Si*) is the most preferred pathway with catalyst combination [Cu(I)/(*S,Sp*)-**L1** + Ir(I)/(*R,R,R*)-**L2**], leads to **Int-3I**, which has (2*S*,3*R*)-configuration (as visualized in Fig. 4, the first quadrant). The

stereochemical outcomes of **Int-3II** (*Re-Si*), **Int-3III** (*Si-Re*), and **Int-3IV** (*Re-Re*) correlated with the other three set of catalyst combinations could be similarly deduced and visualized as shown in the other three quadrants of Fig. 4, respectively. According to the results of the parallel experiments shown in Fig. 3a–d and the confirmed *E*-geometry of the C=C bond and the E/Z-geometry of the C=N bonds in the rearranged imine products in Fig. 3e, f, the stereochemical outcome of the subsequent aza-Cope rearrangement was rationalized as being the result of the stereospecific chirality transfer from the formation of a highly ordered six-membered chair-like transition state (Zimmerman-Traxler model)[52]. Fig. 4 shows the four proposed chair-like transition states (**TS-I-IV**), which are correlated with the four respective intermediates generated with the pairwise catalyst combinations. Since only *E*-olefin was observed in the rearranged final product, only the favored transition states with the 3-phenyl group in an equatorial position were shown for clarification (see SI for other disfavored transition states). The steric repulsion caused by phenyl, ester and isopropyl groups of **Int 3** as well as the relative

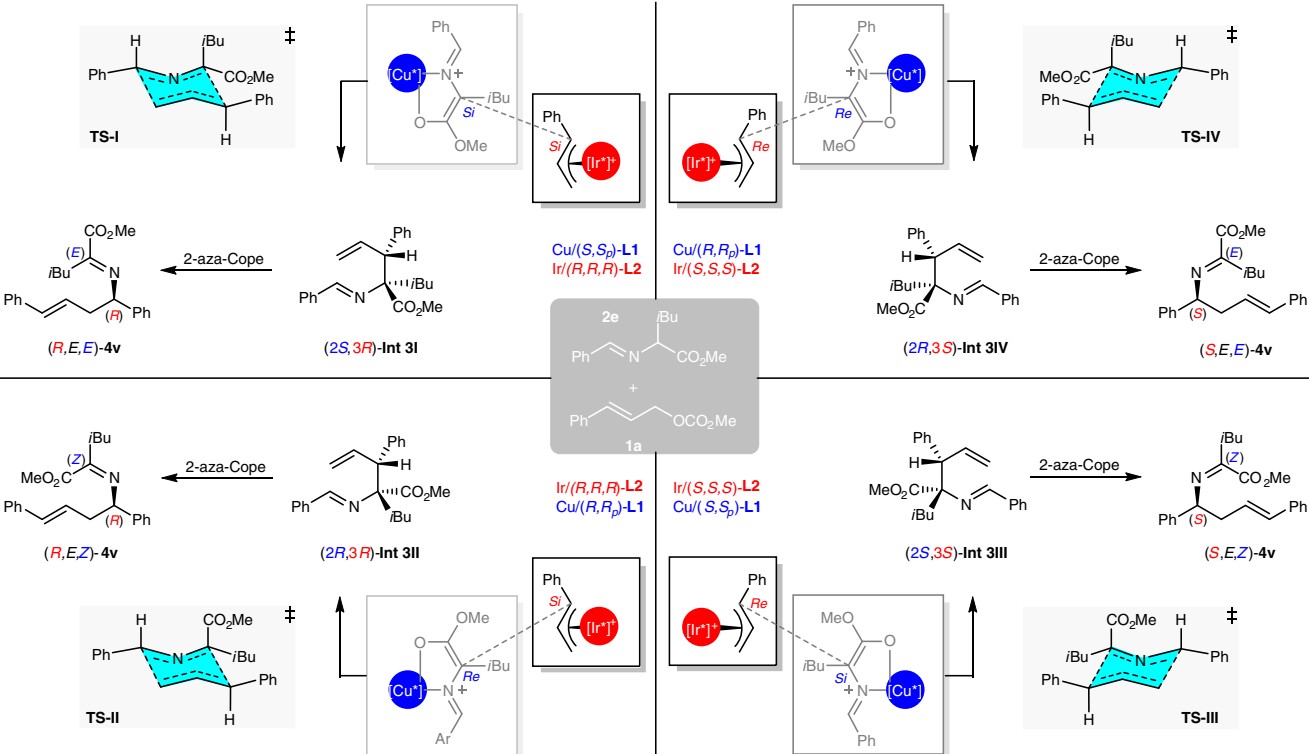

**Fig. 4** Conformational analysis. Different possible approaches between the prochiral faces of Cu(I)-stabilized azomethine ylide and Ir-π-allyl complex in the stereodivergent controlled C–C bond formation and the stereospecific chirality transfer of the 2-aza-Cope rearrangement via the favored chair-like transition states

conjugation-enhanced stability of **4** to **Int 3** provide a sufficient driving force to push the equilibrium of the 2-aza-Cope rearrangement to the product side. The proposed chair-like transition states show that the stereochemical feature of the initially formed **Int 3** is stereospecifically transferred to that of compound **4**. This consistency of the conformational analysis for all the four chair-like transition states further elucidates that the stereogenic center in the rearranged product **4v** was governed by the absolute tertiary allylic stereogenic center in **Int-3**, whereas the geometry of the C=N bond in **4v** was controlled by the relative stereochemistry of the two adjacent stereogenic centers in the initially formed **Int 3**.

According to preliminary mechanistic studies and the results of parallel experiments shown in Fig. 3, we realized that the chirality of copper catalyst is not indispensable for this dual Cu/Ir-catalyzed cascade allylation/2-aza-Cope rearrangement, since two different catalyst combinations ([Cu(I)/(R,Rp)-**L1** + Ir(I)/(S,S,S)-**L2**] and [Cu(I)/(S,Sp)-**L1** + Ir(I)/(S,S,S)-**L2**]) delivered homo-allylic amine (S,E)-**3a** with the same configuration after acidic hydrolysis to obviate the imine moiety (Fig. 3e, f). With [Cu(I)/*rac*-**L1** + Ir(I)/(S,S,S)-**L2**] as the catalyst combination, the cascade reaction proceeded well and delivered expected (S,E)-**3a** in good yield with a comparable level of enantioselectivity (Table 1, entry 1). This result further inspired us to investigate the feasibility of replacing chiral **L1** with certain achiral ligands to further improve the practicality of this method. We then screened a series of commercially available and inexpensive achiral ligands (Table 1, entries 2–9). DPPE, DPPP, DPPB and PPh₃ were not suitable ligands for the copper catalyst, as low conversions of the starting materials were observed. When using DPPF or *rac*-BINAP as the ligand, the cascade reaction proceeded smoothly and furnished desired product (S,E)-**3a** in excellent enantioselectivity and good yield (Table 1, entries 6 and 7). The isolated yield could be further

improved with phenyl ether-linked DPEphos **L3** (Table 1, entry 8). Using this achiral copper complex to activate the aldimine ester, we also examined the behavior of iridium complexes with other privileged phosphoramidite ligands in this reaction[53–55]. It was found that You's ligand[19] (S,Sₐ)-**L4** also exhibited excellent asymmetric induction and catalytic activity (Table 1, entry 9). Control experiments revealed that a copper complex under basic conditions is prerequisite for converting the aldimine ester to the metallated azomethine ylide with a rigid structure and enhanced nucleophilicity. Using a series of inorganic and organic bases in the absence of copper complex significantly retarded the reaction (Table 1, entries 11-15). By comparison, the yield was remarkably increased when Cu(I)/DPEphos was used with Cs₂CO₃ or DBU as the base, which further verified the pivotal role of the copper complex in this cascade reaction (entry 8 *vs* entry 12; entry 14 and entry 16). Thus, with Cs₂CO₃ as the base the combination of [Cu(I)/**L3** + Ir(I)/(S,S,S)-**L2**] was selected as the catalyst combination for the subsequent study. Considering two distinct ligands existed in this reaction, the possible ligand scrambling process was then evaluated by means of ³¹P NMR experiments and control experiments (entries 17 and 18; and also see Supplementary Methods, Mechanistic Study Section for details), the experimental results clearly indicated that the ligand scrambling was negligible or absent and [Cu(I)/**L3** + Ir(I)/(S,S,S)-**L2**] are the active catalysts.

**Substrate scope.** With optimized conditions in hand, we then set out to examine the generality of this protocol. First, a variety of π-allyl precursors were tested with **2c**, and the results are summarized in Fig. 5. Methyl cinnamyl carbonates **1b-1i** with different substituents on the phenyl ring reacted smoothly and gave rise to the corresponding homoallylic amines in good yields with excellent enantioselectivities. Substrates containing electron-

**Table 1 Optimization of reaction conditions**

| Entry | L for Cu | L for Ir | Base | t (h) | Yield (%) | ee (%) |
|---|---|---|---|---|---|---|
| 1 | rac-**L1** | **L2** | Cs₂CO₃ | 6 | 84 | 96 |
| 2 | DPPE | **L2** | Cs₂CO₃ | 24 | <20 | 91 |
| 3 | DPPP | **L2** | Cs₂CO₃ | 24 | <20 | 95 |
| 4 | DPPB | **L2** | Cs₂CO₃ | 24 | <20 | 97 |
| 5 | PPh₃ | **L2** | Cs₂CO₃ | 24 | <20 | N.D. |
| 6 | DPPF | **L2** | Cs₂CO₃ | 6 | 78 | 96 |
| 7 | rac-BINAP | **L2** | Cs₂CO₃ | 6 | 81 | 96 |
| 8 | **L3** | **L2** | Cs₂CO₃ | 6 | 94 | 96 |
| 9 | **L3** | **L4** | Cs₂CO₃ | 6 | 85 | −96 |
| 10 | **L3** | **L5** | Cs₂CO₃ | 12 | 67 | 45 |
| 11 | – | **L2** | Cs₂CO₃ | 24 | Trace | N.D. |
| 12[a] | – | **L2** | Cs₂CO₃ | 24 | Trace | N.D. |
| 13[a] | – | **L2** | LiOtBu | 24 | <20 | 97 |
| 14[a] | – | **L2** | DBU | 24 | 29% | 97 |
| 15[a] | – | **L2** | Et₃N | 24 | Trace | N.D. |
| 16 | **L3** | **L2** | DBU | 12 | 86 | 97 |
| 17 | **L3** | **L3** | Cs₂CO₃ | 12 | – | – |
| 18 | **L2** | **L2** | Cs₂CO₃ | 12 | Trace | N.D. |

*DPPE* 1,2-Bis(di-phenylphosphino)ethane, *DPPP* 1,3-Bis(diphenylphosphino)propane, *DPPB* 1,4-Bis(diphenylphosphino)butane, *Ir(I)* [Ir(COD)Cl]₂, *Cu(I)* Cu(MeCN)₄BF₄
Reactions conditions: **1a** (0.20 mmol), **2c** (0.30 mmol), Cu(I)/**L** (0.01 mmol), Ir(I)/**L** (0.006 mol), CH₂Cl₂ (1 mL), N₂, 25 °C. Isolated yields were shown. Ee was determined by HPLC analysis
[a]Without Cu(I)/DPEphos (**L3**)

neutral, electron-rich or electron-deficient aromatic rings all proved to be viable reaction partners. For *ortho*-substituted cinnamyl carbonates **1j**–**1l**, poor levels of enantioselectivity were observed under the standard conditions, which is a general challenge associated with Feringa-type ligand **L2**. Fortunately, the use of You's phosphoramidite ligand (S,Sa)-**L4** significantly improved the outcome, and desired product **3j**–**3l** were isolated in good yields with 88–97% ee. π-Allyl precursors with a fused aromatic ring (**1m**) or with various heteroaryl groups, including

furan (**1n**), thiophene (**1o**), pyridine (**1p** and **1q**) and quinoline (**1r**) moieties, are well tolerated in this transformation.

When methyl crotyl carbonate **1s** was employed, the reaction was interrupted after the first allylation step without subsequent 2-aza-Cope rearrangement, giving rise to branched intermediate **3s′** in good yield in a 1:1 diastereomeric ratio (99% ee for both isomers). Although the 2-aza-Cope rearrangement was promoted by heating to 100 °C in toluene, the chirality transfer was not fully stereospecific, and the enantioselectivity significantly dropped to

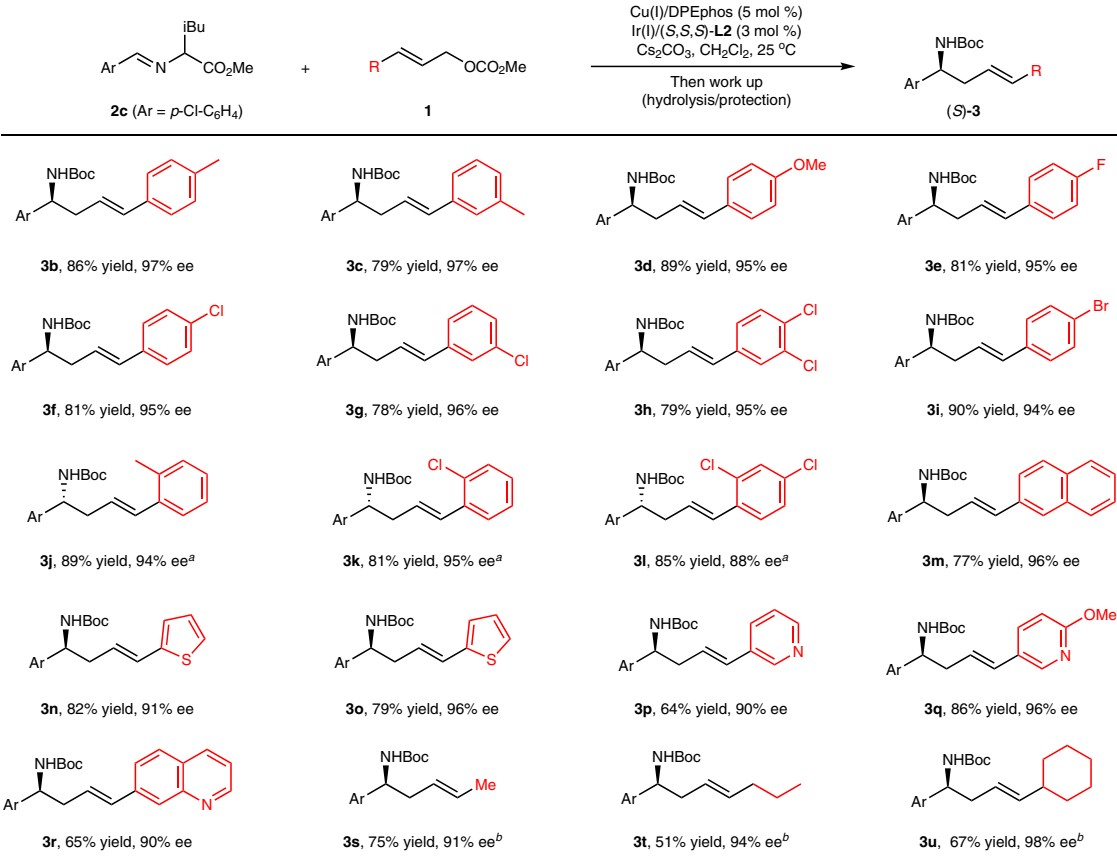

**Fig. 5** Substrate Scope of substituted allyl carbonates. Reactions conditions: **1** (0.20 mmol), **2c** (0.30 mmol), Cu(I)/DPEphos (0.01 mmol), Ir(I)/(S,S,S)-**L2** (0.006 mmol), CH$_2$Cl$_2$ (1 mL), N$_2$, 25 °C, 12–24 h. Isolated yields were shown. Ee values were determined by HPLC analysis. [a][Cu(I)/DPEphos + Ir(I)/(S, S$_a$)-**L4**] complexes were employed as the catalysts. [b][Cu(I)/(S,S$_p$)-**L1** + Ir(I)/(S,S,S)-**L2**] complexes were employed as the catalysts

71% ee. Since the driving force of the 2-aza-Cope rearrangement is mainly from the steric repulsion around the two adjacent stereogenic centers of the allylation intermediate, the slowing of the rearrangement and the reduced efficiency of the chirality transfer with aliphatic allylic carbonates were attributed to the less sterically demanding alkyl group and the absence of additional stabilization from the π–π conjugated system in the final product. This led us to conjecture that further increasing the steric repulsion would facilitate the shifting of the rearrangement equilibrium to the product side and would improve the efficacy of chirality transfer. Based on the quadrant analysis of the favored transition states shown in Fig. 4, we envisioned that a similar level of steric repulsion would be regained as before if α-isobutyl group in aldimine ester **2c** was replaced by a phenyl group, the generated allylation intermediate would undergo the 2-aza-Cope rearrangement via the transition states in the second or third quadrant (the bulkier phenyl group being placed at the equatorial position of the corresponding transition states) more readily than those in the first and the fourth quadrant (the bulkier phenyl group being placed at the axial position of the corresponding transition states). Guided by this conformational analysis, we carried out the reaction of crotyl carbonate **1s** and 2-phenylglycine derived aldimine ester **2d** with the set of matched catalyst combination [Cu(I)/(S,S$_p$)-**L1** + Ir(I)/(S,S,S)-**L2**] (see SI for the detailed experimental results with the set of mismatched catalyst combination [Cu(I)/(R,R$_p$)-**L1** + Ir(I)/(S,S,S)-**L2**]). To our delight, the desired homoallylic amine (S,E)-**3s** could be achieved in good yield with 98% ee upon heating corresponding allylation intermediate (2R,3R)-**Int 3s-III** at 100 °C in toluene for 12 h. With the catalyst combination of [Cu(I)/(S,S$_p$)-**L1** + Ir(I)/(S,S,S)-

**L2**], several other allylic carbonates bearing linear or branched aliphatic groups were well tolerated, and the desired products were furnished in good yields with excellent enantioselectivity upon heating of the corresponding allylation intermediates.

The substrate scope with respect to the aldimine esters was also evaluated (Fig. 6). This protocol shows good tolerance towards both electronic properties and substituent positions when functionalized benzaldehyde derived aldimine esters were used, and these substrates lead to a series of enantioenriched homoallylic amines (**3v**-**3C**) in high yields with 95-98% ee. Aldimine esters bearing an electron-deficient group on the phenyl ring tend to be more reactive than those bearing an electron-rich group. When p-anisaldehyde-derived leucine imine ester **2i** was used, the reaction could be interrupted at the first allylation step. Nevertheless, rearranged homoallylic amine **4z** could be readily generated upon heating the reaction mixture at 50 °C. 1-Naphthyl-, 2-naphthyl- and styrenyl-substituted aldimine esters underwent this transformation smoothly and afforded desired products **3D**-**3E** in 63–89% yields with 95-96% ee. Importantly, a broad array of imine esters originating from hetero-aromatic aldehydes also worked well, giving corresponding products **3F**-**3K** in high selectivity. In addition to the (hetero)aryl-substituted aldimine esters, the current cascade allylation/aza-Cope reaction was also compatible with aldimine esters bearing vinyl and aliphatic groups, such as styrenyl, phenylethyl, isopropyl and cyclohexyl groups.

Since the preliminary study has revealed the stereospecific chirality transfer in 2-aza-Cope rearrangement of a range of the generated allylation intermediates bearing two adjacent quaternary and tertiary stereocenters, we wondered if the efficacy of the chirality transfer could be maintained when the tertiary allyl

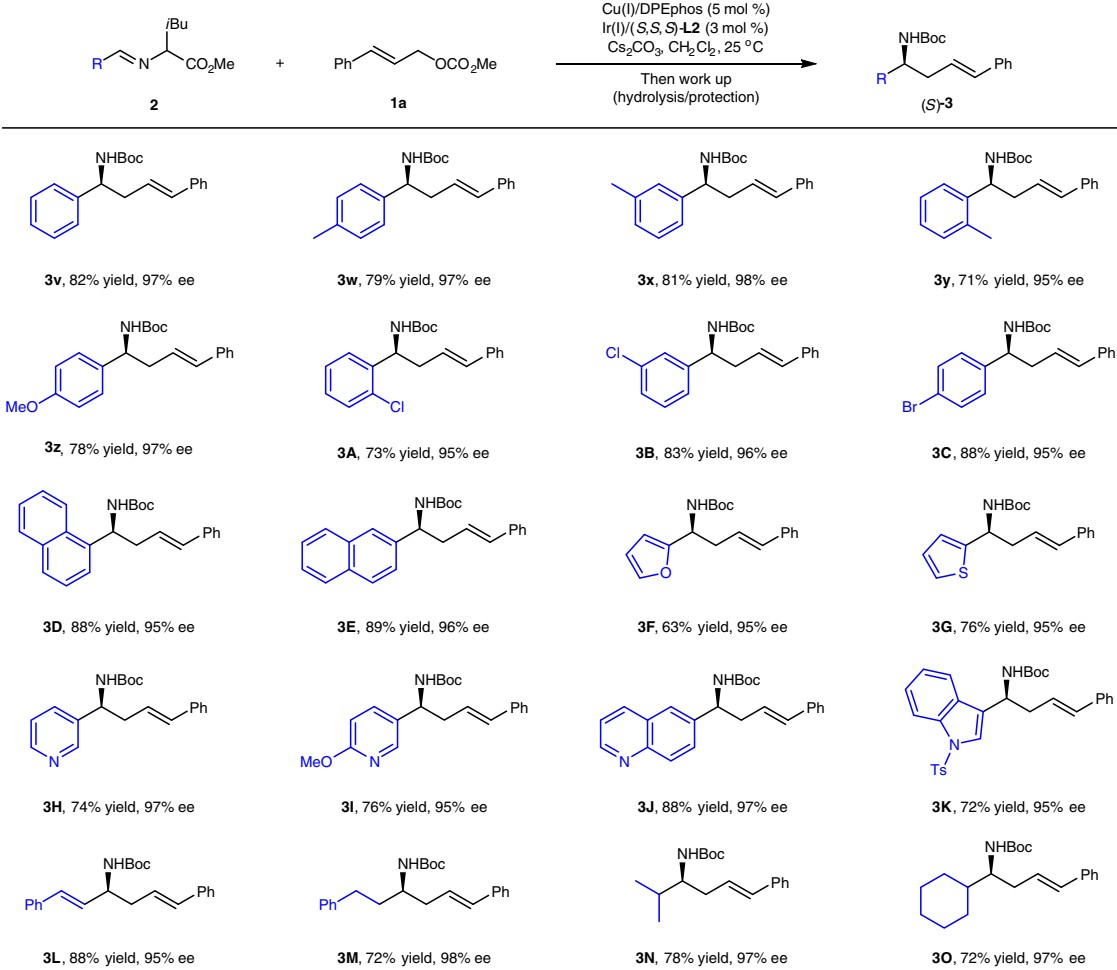

**Fig. 6** Substrate Scope of aldimine esters. Reactions conditions: **1a** (0.20 mmol), **2** (0.30 mmol), Cu(I)/DPEphos (0.01 mmol), Ir(I)/(*S,S,S*)-**L2** (0.006 mmol), CH$_2$Cl$_2$ (1 mL), N$_2$, 25 °C, 12–24 h. Isolated yields were shown. Ee values were determined by HPLC analysis

stereocenter controlled by chiral Ir-catalyst was eliminated in the corresponding allylation intermediates. To further investigate the generality and limitation of this cascade protocol, the reaction of methyl 2-methyl prop-2-en-1-yl carbonate **1v** and phenylglycine derived aldimine ester **2d** was conducted with the combined [Cu (I)/(*R,R$_p$*)-**L1** + Ir(I)/*rac*-**L2**] as the catalyst. Due to the reduced steric congestion, the reaction stopped at the first allylation step at room temperature without further rearrangement, and the corresponding intermediate (*R*)-Int **3P′** was obtained in 99% yield with 97% ee. We were pleased to find that the allylation intermediate (*S*)-Int **3P′** could rearrange smoothly upon heating at 100 °C in toluene for 12 h, and the desired 1,3-disubstituted homoallylic amine (*R*)-**3P** was produced in good yield with 93% ee after work-up. The stereochemistry of this stereospecific chirality transfer could also be similarly rationalized through the quadrant conformational analysis shown in Fig. 4. Further control experiments demonstrated that a comparable level of stereospecific chirality transfer could be achieved with the combined [Cu(I)/(*R,R$_p$*)-**L1** + Ir(I)/(*S,S,S*)-**L2**] or [Cu(I)/(*R,R$_p$*)-**L1** + Ir(I)/(*R,R,R*)-**L2**] as the catalyst, which is consistent with that the absolute configuration of the *N*-quaternary stereogenic center in (*S*)-Int **3P′** is fully controlled by chiral Cu(I)/(*R,R$_p$*)-**L1** complex. More examples with respect to aldimine esters containing electron-neutral, electron-deficient aryl and heteroaryl substituents are further tested, affording the corresponding 1,3-disubstituted homoallylic amine **3Q-T** in good yield with excellent enantioselectivity (Fig. 7). 2-Phenyl allyl carbonate was

also compatible and the desired product **3U** could be obtained in 85% yield with 84% ee. It worth mentioning that there is no report on the catalytic asymmetric construction of enantioen-riched 1,3-disubstituted homoallylic amines except limited examples with the use of moisture-sensitive allylmetal reagents[14]. Remarkably, with bulkier α-naphthalen-1-yl substituted aldimine ester **2y** as the reaction partner, 1-substituted homoallylic amine **3V** could be obtained in moderate yield with 80% ee. These results clearly demonstrated the high efficiency of the stereo-specific chirality transfer of 2-aza-Cope rearrangement even when the allylation intermediate contains only one quaternary stereo-genic center, which further showcase the flexibility of this cascade protocol for the synthesis of diversified homoallylic amines.

**Mechanism explanations and control experiments.** A full mechanism was proposed for this dual Cu/Ir-catalyzed allylation/ aza-Cope rearrangement in Fig. 8. The aldimine ester (**2e**) was activated by the Cu(I)/DPEphos complex to form the achiral Cu-azomethine ylide (**A**) in the presence of base. Meanwhile, the Ir-π-allyl intermediate (**B**) was formed by cinnamyl carbonate (**1a**) and Ir/(*S,S,S*)-**L2** via decarboxylative oxidative addition. The *Si*-face of the Ir-π-allyl intermediate (**B**) was shielded according to the related X-ray structure of the metallacyclic allyl iridium complex in the literature[56]. Therefore, the two catalytic cycles merge via the random approach of either the *Si*-face or the *Re*-face of the achiral ylide (**A**) to the *Re*-face of the chiral Ir-π-allyl intermediate (**B**), which produces the (2*S*,3*S*)- and (2*R*,3*S*)-

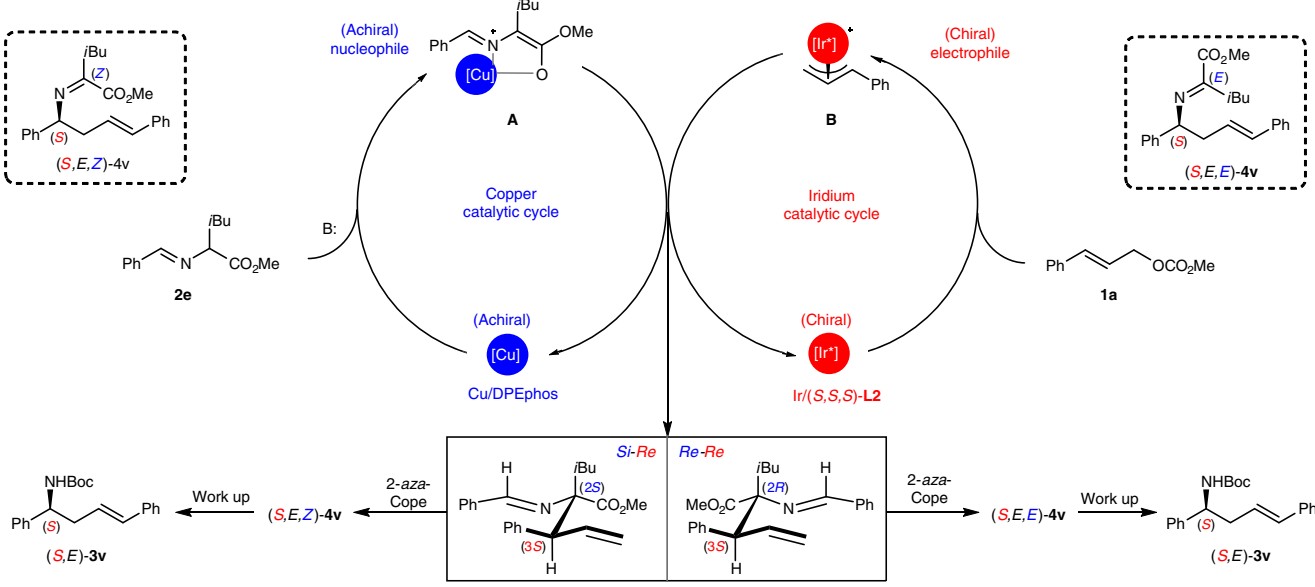

**Fig. 7** Substrate Scope of 2-Substituted Prop-2-en-1-yl Carbonates. Reactions conditions: **1** (0.20 mmol), **2** (0.30 mmol), Cu(I)/(R,$R_p$)-**L1** (0.01 mmol), Ir (I)/rac-**L2** (0.006 mmol), CH$_2$Cl$_2$ (1 mL), N$_2$, 35 °C, 36–48 h. Isolated yields were shown. Ee values were determined by HPLC analysis. [a][Cu(I)/(R,$R_p$)-**L1** + Ir(I)/(S,S,S)-**L2**] complexes were used as the catalyst. [b][Cu(I)/(R,$R_p$)-**L1** + Ir(I)/(R,R,R)-**L2**] complexes were used as the catalyst. [c]α-Naphthalen-1-yl substituted aldimine ester **2y** was used

**Fig. 8** Proposed catalytic cycles and reaction mechanism. Rationale on asymmetric cascade allylation/2-aza-cope rearrangement catalyzed by dual Cu/ DPEphos complex (achiral) and Ir/(S,S,S)-**L2** complex (chiral)

allylation intermediates in an approximately 1:1 ratio. The steric congestion forces the allylation intermediates to proceed through a highly ordered 2-aza-Cope rearrangement to release the steric congestion, and this results in a mixture of Z/E imine isomers **4v** containing the same S stereogenic center and E-geometry of the C=C bond. Finally, hydrolytic work up followed by protection led to the optically active homoallylic amine (S,E)-**3v**.

When p-anisaldehyde and leucine-derived aldimine esters **2i** was used with the catalyst combination [Cu(I)/(S,$S_p$)-**L1** + Ir(I)/ (S,S,S)-**L2**], the branched allylation intermediate could be isolated in good yield as a single isomer, which provided us the opportunity to investigate the roles of the copper and iridium catalysts in the subsequent 2-aza-Cope rearrangement step with

NMR experiments at 50 °C (see the Supplementary Methods, Mechanistic Study Section for details). Control experiments indicated that the Ir complex does not have an observable accelerating effect on the 2-aza-Cope rearrangement. However, the Cu complex did accelerate the subsequent 2-aza-Cope rearrangement. This acceleration effect was attributed to the possible coordination of the copper cation with the imine moiety, which would enhance the electrophilicity of the imine in the branched allylation intermediate and create a more favorable electronic environment for the 2-aza-Cope rearrangement. This conclusion is consistent with the higher reactivity observed with aldimine esters bearing an electron-deficient group on the phenyl ring.

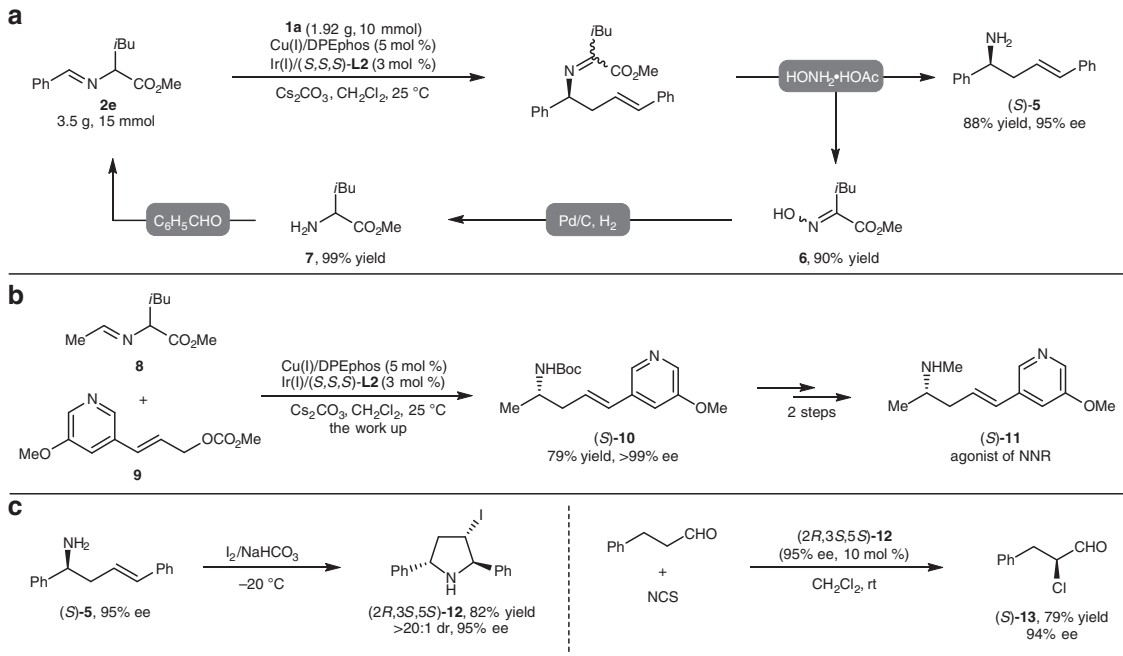

**Fig. 9** Gram Scale and Synthetic Applications. **a** Synthesis of (*S*)-**5** in gram-scale and recycling of by-product. **b** Synthesis of the key intermediate of agonist of NNR. **c** Transformation of (*S*)-**5** to pyrolidine derivatives (2*R*,3*S*,5*S*)-**12** and its application as organocatalyst in asymmetric α-chlorination of alkyl aldehyde

**Gram scale and synthetic applications**. As described in Fig. 4, one equivalent of α-keto ester was formed as the by-product when the rearranged imine was treated with hydrochloric acid. To improve the practicality of this method, we conducted a gram-scale reaction of **1a** and **2e** followed by a modified work-up procedure with hydroxylamine, which delivered desired primary amine (*S*)-**5**[57] with a comparable yield and enantioselectivity (Fig. 9a). Moreover, oxime **6**, isolated in 90% yield, could easily be transformed to *rac*-leucine methyl ester **7** by direct hydrogenation. Reaction between aldimine ester **8** and allylic carbonate **9** under the standard conditions afford corresponding (*S*)-**10** in 79% yield and >99% ee, and this compound is the key intermediate in the synthesis of neuronal nicotinic receptor(NNR) agonist (*S*)-**11**[58] (Fig. 9b). We then investigated the application of the enantioenriched homoallylic amine to show the utility of this protocol. The I₂-promoted cyclization of (*S*)-**5** under basic conditions affords the unprotected *trans*-2,5-diphenylpyrrolidine **12** in good yield with full diastereoselectivity (>20:1 dr) (Fig. 9c, left side); *trans*-2,5-diphenylpyrrolidine is an important building block in the synthesis of chiral ligands[59,60] and chiral auxiliaries[61]. In addition, compound **12** itself is an ideal organocatalyst for promoting the α-chlorination[62] of alkyl aldehydes with excellent asymmetric induction (Fig. 9c, right side).

## Discussion

In conclusion, we have developed a general method for the preparation of a variety of chiral primary homoallylic amines (1,4- and 1,3-disubstituted homoallylic amines) through a synergistic Cu/Ir-catalyzed cascade allylation/2-aza-Cope rearrangement process. A broad range of substrates bearing various functional groups were tolerated in this reaction and gave the corresponding homoallylic amines in high yields with excellent enantioselectivities even on a large scale. More significantly, the mechanism study clearly revealed the distinct transition states of the aza-Cope rearrangement from the diastereomeric allylation intermediates, and the flexible role of the tertiary/quaternary stereogenic centers in the chirality transfer and the geometry control of the C=C and C=N bonds was elucidated. Finally, the utility of the current methodology has been proven by a series of synthetic transformations. We expect this method will inspire the development of synergistic catalysis and further investigations of the mechanism of [3,3]-sigmatropic rearrangement.

## Methods

**General reaction procedure A**. A flame dried Schlenk tube I was cooled to rt and filled with N₂. To this flask were added [Ir(COD)Cl]₂ (0.003 mmol, 1.5 mol %), phosphoramidite ligand (*S,S,S*)-**L2** (0.006 mmol, 3 mol %), degassed THF (0.5 mL) and degassed *n*-propylamine (0.5 mL). The reaction mixture was heated at 50 ºC for 30 min and then the volatile solvents were removed under vacuum to give the iridium complex as a pale yellow solid. Meanwhile, in a separated Schlenk tube II, DPEphos (0.011 mmol, 5.5 mol %) and Cu(MeCN)₄BF₄ (0.01 mmol, 5 mol %) were dissolved in 0.5 mL of CH₂Cl₂, and stirred at room temperature for about 0.5 h. The Cu/DPEphos complex solution was then transferred to the Schlenk tube I containing iridium complex via syringe. Allylic carbonate (0.20 mmol), leucine derived aldimine ester (0.30 mmol), Cs₂CO₃ (0.30 mmol) and CH₂Cl₂ (0.5 mL) were then added. The cascade reaction was finished smoothly in 12–24 h (monitored by ¹H NMR spectroscopy). Then, 2N HCl (0.5 mL) was added to the mixture. After stirring for 0.5 h, 2N NaOH (1 mL) and Boc₂O (88 mg, 0.4 mmol) was added and stirred at rt for 3 h. The layers were separated, and the aqueous layer was extracted with CH₂Cl₂ (5 mL × 2). The combined organic components were washed with saturated brine (10 mL), dried over anhydrous Na₂SO₄, filtration and evaporated in vacuum. The residue was purified by column chromatography to give the desired product, which was then directly analyzed by HPLC to determine the enantiomeric excess.

## Data availability

Experimental procedures and characterization data are available within this article and its Supplementary Information. Data are also available from the corresponding author on request.

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

## Acknowledgements

This work was supported by NSFC (21525207, 21772147), China Postdoctoral Science Foundation funded project (2017M620331). The Program of Introducing Talents of Discipline to Universities of China (111 Program) is also appreciated. We thank Prof. Shu-Li You at Shanghai Institute of Organic Chemistry for generously providing ligand (*S*,*S*$_a$)-**L4**.

## Author contributions

C.J.W. and L.W. conceived and designed the research. L.W., Q.Z., L.X., and H.Y.T. performed the research. C.J.W. and L.W. co-wrote the paper. All authors analyzed the data, discussed the results, and commented on the manuscript.

## Additional information

**Competing interests:** The authors declare no competing interests.

