## [Peer Review File · Nature Communications]

Reviewers' comments:

Reviewer #1 (Remarks to the Author):

In this paper, Wang et al. report the “Synergistic Cu/Ir Catalyzed Asymmetric Cascade Allylation/2-aza-Cope Rearrangement of Azomethine Ylides”. The results presented herein build upon previous results from the same group regarding Cu/Ir dual catalysis for the stereodivergent construction of α,α -disubstituted α -amino acids by the asymmetric allylation of aldimine esters with full control of the diastereoselectivity and enantioselectivity. In this work, the authors elaborate on the in situ aza-cope rearrangement of putative quaternary imine intermediates to yield homoallylic amines with excellent yields and enantioselectivities. While the experimental reaction optimization (screening of different ligands for the metal catalysts and base) and substrate scope examination is remarkable, the computational mechanistic study and the way it is presented lacks the necessary thoroughness to be published in Nature Communications. These are some major issues the authors must solve before the whole work is acceptable for publication:

- Strikingly, the absolute energies of the reported structures in Hartree are huge, and span in the range of -688917.5585881 to -688880.1031378. When translated to kcal mol⁻¹, the relative energies go up to 23503 kcal mol⁻¹ !!! This referee repeated the calculations at a comparable level of theory and the energies are around -1096.6 Hartree. Clearly, something is wrong with the numbers given by the authors.
- A more detailed explanation of how the relative energies (in kcal mol⁻¹) are calculated for each presented stationary point must be provided. Apparently, single-point energies were calculated with M06-2X/def2-TZVP using the geometries optimized with M06-2X/def2-SVP, but it is not clear at which stage(s) did the authors include empiric dispersion (GD3?, GD3BJ? Grimme's program?) and solvent corrections.
- The energies of the vast majority of the stationary points are omitted in the cartesian coordinates section, precluding the validation of the relative energies presented in Figures S2-S14. This is not acceptable.
- Along these lines, and for reproducibility purposes, a detailed table summarizing the electronic energies, zero-point corrected energies, enthalpies, free energies, entropies, number of negative frequencies and the value of the lowest frequencies for each presented stationary point, must be provided in the Supporting Information.
- Labels for all the calculated structures must be provided throughout Figures S1-S14, so they can be identified later on in the energies (to be added) and coordinates sections.
- It is not clear to which extent did the authors explore different conformations for each reported stationary point. This referee has performed a somewhat systematic (but still not complete) conformational analysis on some of the structures presented in Figures S1 and S2 (using

the provided cartesian coordinates and with m062x/def2SVP scf=tight int=ultrafine opt freq=noraman scrf=(smd,solvent=dichloromethane), using Gaussian 16) and has found multiple lower energy conformers with free energies up to 2 kcal mol⁻¹ smaller than the presented structures. This is particularly important for ChairTS-2R3R-REZ (Figure S3), which happens to be around 2 kcal mol⁻¹ lower than the structure reported, and yields an activation equal or even lower than that of ChairTS-2S3R-REE (Figure S1). This must be fixed, and the authors should make sure that absolutely all possible conformations are calculated and available to an external reader.

- The authors assume that the aza-Cope rearrangement is thermal, i.e. without the assistance of any of the transition metal catalysts present in the reaction mixture. This, of course, simplifies the calculations but might not reflect reality. Particularly, I don't find a strong reason for the copper atom not to be coordinated at the aldimine N after the α -allylation of the ester enolate. At least in one of the cases for which the imine intermediate is detected, the thermal reaction (i.e. after work-up and purification, or at least removal of the catalysts) must be assayed. A careful discussion on this issue is necessary.

- The way the computational results are presented in Figures S1-S14 is confusing and not accurate. The authors consider the "chair" and "boat" conformers to be somewhat disconnected energetically, and activation energies are presented with respect to each particular conformer (two blue and two black pathways in two different graphs), which violates the Curtin-Hammett principle. Such principle involves that the relative energies and associated activation barriers must be derived using the same lowest-energy minimum as a reference. Hence, and since Figures S1-S2 describe different reaction pathways involving different conformations of the same intermediate, a more detailed plot depicting all pathways and the interconnection between the different stationary points must be provided.

- No three-dimensional representations of the calculated structures are presented, which is really staggering and makes the computational study impossible to follow. The authors should add pictures of the optimized structures for all the calculated stationary points, highlighting important distances and dihedral angles to illustrate torsional strain, etc.

It is not surprising that the configuration at the 2-position determines the stereochemistry of the aza-Cope rearrangement using simple three-dimensional modeling. The more stable and conformationally relaxed chair-like transition structure (Zimmerman-Traxler model), will always locate the aryl group at the 2-position (allylbenzene) in an extended and equatorial arrangement. In turn, the E configuration of the initial aldimine ester also forces its aryl substituent to adopt an equatorial orientation at the chair-like TS. Thus, this mutually equatorial disposition of the aryl groups will determine the configuration of the resulting homoallylic amine. In order to access the corresponding epimer, at least one of the two aryl groups must adopt an axial disposition and a folded conformation of the allylbenzene moiety, which significantly distorts the geometry of the chair-like TS and increases the activation barrier. The configuration at the 3-position is quite irrelevant since it is a quaternary carbon with groups with relatively similar sizes, as reflected by the TS activation energies (nearly identical as re-calculated by this referee). The computational study should reflect these basic concepts of stereoiduction instead of focusing so insistently on the importance of the size of the substituent at the quaternary carbon, in a clear and comprehensible

manner and with the assistance of three-dimensional optimized structures. One of the major goals of computational studies is to explain experimental results and make them comprehensible, not just agree with experiments and complicate the explanations.

Minor issues:

- I would consider that the initial step of the discussed reaction is an α -allylation of an ester enolate rather than of an azomethine ylide, since the aldimine is not formally an iminium.
- The charges in Scheme 6 and in other sections throughout the manuscript should be represented in a more consistent and chemically reasonable way. For instance, the negative charge after aldimine deprotonation should be better located at the oxygen of the corresponding enolate, which furthermore coordinates to the copper atom. In the rest of the quadrants of this figure the charges disappear, why? Also, the positive charge of the Ir-allyl cation should be properly represented, or the positive charge at the N of the azomethine ylide removed since it arises from coordination to copper...
- In line 149 of the manuscript it is stated that "(S,E,Z)-4q, containing Z-geometry of imine moiety, was produced as the major isomer", but in Scheme 5, the Z/E isomer ratio is presented as Z/E = 1:11, indicating a large prevalence of the E isomer. This must be corrected.
- In the captions of Figures S1-S4, mentions to "Figure 7" must be corrected to "Figure 6".
- In line 282 "we wondered the efficacy" should be corrected to "we wondered if the efficacy".
- In lines 295-296 "which is consistence with" should be corrected to "which is consistent with"

Reviewer #2 (Remarks to the Author):

The present manuscript describes the allylation of iminoester derivatives via the Cu- and Ir-catalysis, in which the nucleophilic addition of a Cu-enolate intermediate to an allyliridium intermediate takes place and the subsequent aza-Cope rearrangement gives the desired homoallylic amine derivatives with high to excellent enantioselectivities. The sterically congested intermediates can promote the aza-Cope rearrangement under ambient conditions. The examination of a chirality transfer from an alpha-allylester intermediate showed the influence of each chiral center at alpha and beta-positions. The present reaction for a formal allylation of imine derivatives seems to be somehow verbose, but

the rationale of the present approach via the dual catalyst systems and the information of following rearrangement would be valuable for the interest of readers of Nature Communications. The reviewer recommends the publication in Nature Communications after minor revision.

In Table 1, the possibility of ligand exchange between a Cu and a Ir-complex should be described to elucidate the exact role of chiral ligands.

The authors suggest that the sterically congested intermediates could promote aza-Cope rearrangement. In this context, the authors should address substrate 2 (R = H) to the tandem allylation and aza-Cope rearrangement even at high reaction temperature. The results in Schemes 4 and 5 indicate that substituent R might not be important for the asymmetric induction for final products, in some cases.

Reviewer #3 (Remarks to the Author):

In this manuscript, Wang and coworkers reported a general method for the preparation of synthetically important, chiral 1,4- and 1,3-disubstituted homoallylic amines through a synergistic Cu/Ir-catalyzed allylation/2-aza-Cope rearrangement cascade process. This reaction exhibited remarkable substrate scope, and displayed excellent enantio-control for most substrates. The authors also demonstrated the utility of these products as precursors to chiral organocatalysts.

The successful development of this transformation is hinged upon very careful mechanistic analysis. The authors first found the steric bulkiness of the R' group in the iminoester substrate is critical for the efficiency of the initial allylation step and the subsequent rearrangement. With this information, they identified that iso-butyl group and phenyl group are optimal substituents. The authors later determined that only the tertiary center is important for the stereochemistry of the final products. (Notably, the authors could make this observation because their catalytic system could furnish both diastereoisomers cleanly.) With this information, they reasoned that this reaction could be performed using achiral ligands to copper. Their following studies supported their own hypothesis, and made the reaction even more practical. Lastly, the authors performed very careful studies to understand that the copper catalyst used in this system could accelerate the aza-Cope rearrangement. This result by itself is also interesting and should inspire the development of other reactions capitalizing on this underexplored aza-Cope rearrangement.

With the above said, I believe this work by the Wang group is of both synthetic value and significant mechanistic insights. Moreover, the SI is of high quality. I therefore highly recommend its publication in Nature Communications.

First of all, we want to thank all the reviewers for their useful and productive comments, which helped to improve the clarity and quality of this manuscript. After reading the comments, we have supplemented several experiments in last several days. Now, we are sure that we can address all the questions raised by the referees. We have tried to revise the manuscript in line with the suggestions made by the referees, and the revisions have been highlighted in yellow.

In the original manuscript, the DFT calculations were conducted to support the experimental results on stereochemistry of 2-aza-Cope rearrangement. We believe the reaction mechanism could be rationally elucidated by the original experimental results and the supplemental control experiments. **Thus, we determined to take the alternative suggestion from the associate editor Dr. Giovanni Bottari and delete the DFT calculations in the revised manuscript. However, we still take the comments other than DFT calculations raised by Reviewer #1 seriously (*vide infra*).**

We are truly grateful for all the productive comments raised by Reviewer #1. It is worth mentioned that Reviewer #1 also agreed with the transition states we proposed in the original manuscript based on the control experiment to explain the stereochemistry of 2-aza-Cope rearrangement (see the corresponding comments: *“It is not surprising that the configuration at the 2-position determines the stereochemistry of the aza-Cope rearrangement using simple three-dimensional modeling. The more stable and conformationally relaxed chair-like transition structure (Zimmerman-Traxler model), will always locate the aryl group at the 2-position (allylbenzene) in an extended and equatorial arrangement. In turn, the E configuration of the initial aldimine ester also forces its aryl substituent to adopt an equatorial orientation at the chair-like TS. Thus, this mutually equatorial disposition of the aryl groups will determine the configuration of the resulting homoallylic amine. In order to access the corresponding epimer, at least one of the two aryl groups must adopt an axial disposition and a folded conformation of the allylbenzene moiety, which significantly distorts the geometry of the chair-like TS and increases the activation barrier”*).

For Reviewer 1’s comments other than DFT calculations:

Q1: *The authors assume that the aza-Cope rearrangement is thermal, i.e. without the assistance of any of the transition metal catalysts present in the reaction mixture. This, of course, simplifies the calculations but might not reflect reality. Particularly, I don’t find a strong reason for the copper atom not to be coordinated at the aldimine N after the α -allylation of the ester enolate. At least in one of the cases for which the imine intermediate is detected, the thermal reaction (i.e. after work-up and*

purification, or at least removal of the catalysts) must be assayed. A careful discussion on this issue is necessary.

Answer 1: The acceleration effect of 2-aza-Cope rearrangement has been carefully discussed in the original manuscript and Supplemental Information based on the control experiments (See revised Supplemental Information, pages 36-41). For your convenience, we copied the corresponding context as below:

When *p*-anisaldehyde and leucine-derived aldimine esters **2i** was used with the catalyst combination [Cu(I)/(*S,S*_p)-**L1** + Ir(I)/(*S,S,S*)-**L2**], the branched allylation intermediate could be isolated in good yield as a single isomer, which provided us the opportunity to investigate the roles of the copper and iridium catalysts in the subsequent 2-aza-Cope rearrangement step with NMR experiments at 50 °C (see Supplemental Information for details). Control experiments indicated that the Ir complex does not have an observable accelerating effect on the 2-aza-Cope rearrangement. However, the Cu complex did accelerate the subsequent 2-aza-Cope rearrangement. This acceleration effect was attributed to the possible coordination of the copper cation with the imine moiety, which would enhance the electrophilicity of the imine in the branched allylation intermediate and create a more favorable electronic environment for the 2-aza-Cope rearrangement. This conclusion is consistent with the higher reactivity observed with aldimine esters bearing an electron-deficient group on the phenyl ring.

Q2: *I would consider that the initial step of the discussed reaction is an α -allylation of an ester enolate rather than of an azomethine ylide, since the aldimine is not formally an iminium.*

Answer 2: ‘Metallated azomethine ylide’ and its representative structure (see Scheme 1 of our current manuscript) have been commonly used in literatures, including both review articles and computational study-involved research articles. For examples, Scheme 1 in *Chem. Commun.* **2014**, 50, 12434; Scheme 77 in *Chem. Rev.* **2015**, 115, 5366; Scheme 1 in *Angew. Chem. Int. Ed.* **2006**, 45, 1979, and so on.

For your convenience, I copied the schemes from above-mentioned literatures as below:

Scheme 1 Classical catalytic asymmetric 1,3-dipolar cycloaddition using α -iminoesters as dipole precursors.

(copied from *Chem. Commun.* **2014**, 50, 12434)

Scheme 77. 1,3-DC of *N*-Metalated Azomethine Ylides

(copied from *Chem. Rev.* **2015**, 115, 5366)

Scheme 1. Proposed mechanism of the reaction of 1 with 2.

(copied from computational study-involved article *Angew. Chem. Int. Ed.* **2006**, 45, 1979)

Q3: The charges in Scheme 6 and in other sections throughout the manuscript should be represented in a more consistent and chemically reasonable way. For instance, the negative charge after aldimine deprotonation should be better located at the oxygen of the corresponding enolate, which furthermore coordinates to the copper atom. In the rest of the quadrants of this figure the charges disappear, why? Also, the positive charge of the Ir-allyl cation should be properly represented, or the positive charge at the N of the azomethine ylide removed since it arises from coordination to copper...

Answer 3: As suggested, the charges of Cu-coordinated azomethine ylide and the positive charge of the Ir-allyl cation have been correctly presented in a more consistent and chemically reasonable way in the revised manuscript.

Q4: *In line 149 of the manuscript it is stated that “(S,E,Z)-4q, containing Z-geometry of imine moiety, was produced as the major isomer”, but in Scheme 5, the Z/E isomer ratio is presented as Z/E = 1:11, indicating a large prevalence of the E isomer. This must be corrected.*

Answer 4: As suggested, the mistake in Scheme 5 about the Z/E ratio has been corrected in the revised manuscript.

Q5: *In line 282 “we wondered the efficacy” should be corrected to “we wondered if the efficacy”*

Answer 5: As suggested, the sentence has been rewritten in the revised manuscript.

Q6: *In lines 295-296 “which is consistence with” should be corrected to “which is consistent with”*

Answer 6: As suggested, the mistake have been corrected in the revised manuscript.

For Reviewer 2:

Q1: *In Table 1, the possibility of ligand exchange between a Cu and a Ir-complex should be described to elucidate the exact role of chiral ligands.*

Answer 1: Two different types of ligand (**L1** and DPEphos **L3**) were used for the formation of chiral or achiral copper complexes in this reaction. For ligand scrambling experiments between Cu/(*S,S*_p)-**L1** and Ir/(*R,R,R*)-**L2**, please see our previous report (Supporting Information of *J. Am. Chem. Soc.* **2018**, 140, 1508). Therefore, further ³¹P NMR experiments have been carried out to investigate the possibility of ligand exchange between Cu/DPEphos(**L3**) and iridium complex, and the corresponding experimental results have been supplemented in the revised manuscript and the revised Supplemental Information (see: pages S42-S44).

Based on the results of ³¹P NMR experiments and control experiments, it is believed that the ligand scrambling was negligible or absent and the combined copper and iridium complexes are the active catalysts.

Q2: The authors suggest that the sterically congested intermediates could promote aza-Cope rearrangement. In this context, the authors should address substrate 2 ($R = H$) to the tandem allylation and aza-Cope rearrangement even at high reaction temperature. The results in Schemes 4 and 5 indicate that substituent R might not be important for the asymmetric induction for final products, in some cases.

Answer 2: We did try aldimine esters with various substituents to evaluate the steric effect in the 2-aza-Cope rearrangement, the results are summarized in Table S1 of the revised Supplemental Information (see: pages 41-42) and also copied as below:

The screening of aldimine esters with different substituents strongly indicated that the steric effect is the key fact for 2-aza-Cope rearrangement (Table S1). For instance, aldimine esters with less bulky α -substituents such as benzyl and methyl group provide the allylated intermediates without sufficient steric congestion, and therefore subsequent 2-aza-Cope rearrangement proceeded slowly. On the other hand, for glycine derived aldimine ester **2C**, the corresponding allylated intermediate was unstable at high temperature, and most of the intermediate decomposed in toluene at 110 °C.

Table S1. Initial Test to Find an Aldimine Ester Bearing an Appropriate α -Substituent to Facilitate Both Allylation and 2-aza-Cope Rearrangement.

Entry ^a	R	allylation	T for 2-aza-Cope/°C	2-aza-Cope	yield (%) ^b	ee (%) ^c
1	tert -Butyl (2a)	×	25	-	-	-
2	iso -Propyl (2b)	×	25	-	-	-
3	iso -Butyl (2c)	√	25	√	86	96
4	Phenyl (2d)	√	25	√	83	95
5	Benzyl (2A)	√	110	√	12	-
6	Methyl (2B)	√	110-150	×	-	-
7	H (2C)	√	110	decomposed	-	-

^a All reactions were carried out with 0.20 mmol of **1a** and 0.30 mmol of **2** in 1 mL of CH₂Cl₂. ^b Isolated yield. ^c Ee was determined by HPLC analysis.

REVIEWERS' COMMENTS:

Reviewer #2 (Remarks to the Author):

The reviewer appreciates the reply from the authors and revisions. The present revised one would be suitable for publication.